# Comparative Transcriptomic Analysis of Archival Human Vestibular Schwannoma Tissue from Patients with and without Tinnitus

**DOI:** 10.3390/jcm12072642

**Published:** 2023-04-01

**Authors:** Krishna Bommakanti, Richard Seist, Phanidhar Kukutla, Murat Cetinbas, Shelley Batts, Ruslan I. Sadreyev, Anat Stemmer-Rachamimov, Gary J. Brenner, Konstantina M. Stankovic

**Affiliations:** 1Department of Otolaryngology–Head and Neck Surgery, Massachusetts Eye and Ear and Harvard Medical School, Boston, MA 02114, USA; 2Department of Head and Neck Surgery, University of California Los Angeles, Los Angeles, CA 90095, USA; 3Department of Otolaryngology–Head and Neck Surgery, Stanford University School of Medicine, Stanford, CA 94305, USA; 4Department of Otorhinolaryngology–Head and Neck Surgery, Paracelsus Medical University, 5020 Salzburg, Austria; 5Department of Anesthesia, Critical Care and Pain Medicine, Massachusetts General Hospital and Harvard Medical School, Boston, MA 02114, USA; 6Department of Molecular Biology, Massachusetts General Hospital and Harvard Medical School, Boston, MA 02114, USA; 7Department of Pathology, Massachusetts General Hospital and Harvard Medical School, Boston, MA 02114, USA; 8Department of Neurosurgery, Stanford University School of Medicine, Stanford, CA 94305, USA; 9Wu Tsai Neuroscience Institute, Stanford University, Stanford, CA 94305, USA

**Keywords:** vestibular schwannoma, tinnitus, inflammation, bioinformatic analysis, gene set enrichment analysis, functional enrichment analysis

## Abstract

Vestibular schwannoma (VS) is an intracranial tumor that commonly presents with tinnitus and hearing loss. To uncover the molecular mechanisms underlying VS-associated tinnitus, we applied next-generation sequencing (Illumina HiSeq) to formalin-fixed paraffin-embedded archival VS samples from nine patients with tinnitus (VS-Tin) and seven patients without tinnitus (VS-NoTin). Bioinformatic analysis was used to detect differentially expressed genes (DEG; i.e., ≥two-fold change [FC]) while correcting for multiple comparisons. Using RNA-seq analysis, VS-Tin had significantly lower expression of *GFAP* (logFC = −3.04), *APLNR* (logFC = −2.95), *PREX2* (logFC = −1.44), and *PLVAP* (logFC = −1.04; all *p* < 0.01) vs. VS-NoTin. These trends were validated by using real-time RT-qPCR. At the protein level, immunohistochemistry revealed a trend for less PREX2 and apelin expression and greater expression of NLRP3 inflammasome and CD68-positive macrophages in VS-Tin than in VS-NoTin, suggesting the activation of inflammatory processes in VS-Tin. Functional enrichment analysis revealed that the top three protein categories—glycoproteins, signal peptides, and secreted proteins—were significantly enriched in VS-Tin in comparison with VS-NoTin. In a gene set enrichment analysis, the top pathway was allograft rejection, an inflammatory pathway that includes the *MMP9, CXCL9, IL16, PF4, ITK,* and *ACVR2A* genes. Future studies are needed to examine the importance of these candidates and of inflammation in VS-associated tinnitus.

## 1. Introduction

Vestibular schwannoma (VS), a tumor arising from neoplastic Schwann cells of the vestibular branch of cranial nerve VIII, is the most common tumor of the cerebellopontine angle and the fourth most common intracranial tumor [1,2]. The majority (95%) of VS cases are unilateral and occur spontaneously in patients without a family history, although bilateral VS is a feature of neurofibromatosis type 2 (NF2), a rare autosomal dominant disorder [2,3,4]. The most common presenting symptoms are sensorineural hearing loss and tinnitus ipsilateral to the VS tumor [5,6]. While histologically benign and typically slow-growing, VS can cause complications due to the tumor’s common location within the internal auditory canal [7,8]. If left untreated, growing VS tumors may cause cranial nerve paralysis, ataxia, hydrocephalus, and even death due to brainstem compression [2,9,10]. Accordingly, the mainstay treatment for growing VS is surgical tumor resection or stereotactic radiotherapy [2,11,12,13,14].

Tinnitus, the perception of sound without an external stimulus, affects approximately 10% of adults in the United States [15], but is present in >70% of patients with VS [16,17,18,19]. Tinnitus can be highly detrimental to patients’ quality of life and mental health, particularly for those with chronic bothersome symptoms [20,21]. The precise mechanism by which VS is implicated in tinnitus—or why certain patients with VS develop tinnitus while others do not—is currently unknown [18,22]. The scientific consensus is that tinnitus is likely a symptom encoded in the central nervous system but is triggered by cochlear dysfunction [23,24,25,26], which could, in turn, be prompted by mechanical compression or the secretion of ototoxic or pro-inflammatory factors by the VS tumor. However, a common clinical observation is that VS tumor size is not correlated with the degree of a patient’s hearing loss [27], which is the main risk factor for tinnitus in VS [22]. Additionally, tinnitus symptoms do not necessarily resolve after VS tumor resection [16,28]. The involvement of secreted immune factors is bolstered by recent studies implicating inflammation in the microenvironment of VS tumors associated with poor hearing and by observations of cochlear cell damage in animal models of tinnitus [29,30]. Additionally, contralateral auditory dysfunction is present or develops in many patients with unilateral VS, which is suggestive of systemic involvement via tumor-released ototoxic factors [31].

A comparison of RNA sequencing profiles revealed differential and higher expression of FGF7 in painful versus non-painful schwannomas among NF2 patients, thus identifying a potential mediator of pain [32]. Similarly, examining differentially expressed genes in the VS tumors of patients with or without tinnitus may help identify pathways or genes that are comparatively enriched in the presence of tinnitus. While tinnitus in the general population is known to have heterogeneous etiologies, those with VS and tinnitus are more alike in pathophysiology, symptoms, and signs than the overall population [18,33]. Thus, uncovering the molecular and genetic components of tinnitus is facilitated in a more homogenous VS population. Accordingly, the aim of this cohort study was to examine whether there was differential gene expression when comparing archival formalin-fixed paraffin-embedded VS tumor tissue from patients with versus without presurgical tinnitus by using next-generation sequencing.

## 2. Methods

### 2.1. Study Population and Tissue Collection

The overall methodological approach of this study is illustrated in Figure 1. Formalin-fixed paraffin-embedded (FFPE) VS specimens from 19 anonymized adults who underwent surgical VS resection at the Massachusetts General Hospital (MGH) or Massachusetts Eye and Ear (MEE) were received from the hospitals’ neuropathology tumor banks. Two patients had NF2-related VS, while 17 had sporadic VS. The specimens were handled according to a study protocol approved by the institutional review board of the Human Studies Committees of MGH and MEE (Protocol # 14-148H).

Demographic (age and sex) and clinical information were extracted from patients’ medical charts. The clinical information included presurgical tumor size measured via magnetic resonance imaging, word recognition percentage (WR%), pure tone average (PTA) decibel (dB) level, and history of tinnitus (i.e., clinical diagnosis prior to resection). Tumor size, PTA dB level, and WR% were based on the most recent tests/scans prior to surgical resection. VS tumor samples were categorized as being from patients with or without presurgical tinnitus based on their history of tinnitus, defined as a diagnosis of tinnitus made by a clinician or audiologist at the time of tumor resection. Tinnitus evaluations were conducted by clinicians experienced in diagnosing inner ear disorders. As tinnitus is an inherently subjective disorder, diagnosis primarily relies on patient self-reporting, patient history, audiometric measures, detailed inquiry of the impact and duration of tinnitus, tinnitus matching, and neuropsychological assessment [34,35].

### 2.2. RNA Extraction and Quality Assessment

Hematoxylin and eosin (H&E)-stained slides of each VS tumor were reviewed by an experienced neuropathologist (ASR). Areas (blocks) enriched for tumor tissue without inflammation were identified to enrich for schwannoma cells and not infiltrating inflammatory cells. Regions of interest were marked, and 6–10 slides from the same block were requested for each sample for RNA extraction. Slides were deparaffinized with xylene and rehydrated as previously described [32]. Six to 10 slides from each patient sample were stained with cresyl violet to identify neural cells and macro-dissected within the region of interest identified on the H&E counterparts.

Total RNA was purified by using RNeasy spin columns (Qiagen, Valencia, CA, USA) per the manufacturer’s protocol. Quantification and quality assessment of the purified RNA were performed at the Dana Farber Cancer Institute (DFCI) Core Facility by using an Agilent 2100 Bioanalyzer and RNA Pico Kit (Agilent Technologies, Santa Clara, CA, USA). Samples were considered of sufficient quality if the percentage of RNA fragments was >200 nucleotides ([DV200] > 20%), an acceptable range for the sequencing of FFPE samples per the DFCI’s recommendations [36].

### 2.3. Complementary DNA (cDNA) Library Preparation and RNA Sequencing (RNA-Seq)

cDNA library preparation and RNA-seq were performed by the DFCI Core Facility. Because of the expected degradation of RNA from the FFPE tissue, ribosomal RNA was removed by using the RiboZero rRNA Removal Kit (Epicentre Biotechnologies, Madison, WI, USA) prior to cDNA library preparation. Double-stranded cDNA libraries were prepared by using the KAPA HyperPrep Kit (Roche, Basel, Switzerland) while following the manufacturer’s protocol. KAPA cDNA library quality control was performed prior to pooling libraries for flow cell amplification. All cDNA libraries were sequenced by using Illumina HiSeq 2000 (Illumina, San Diego, CA, USA) to produce 50 million reads—50 base-pair (bp) paired-end reads with multiplexing (12 samples/lane). RNA-seq was performed with an Illumina HiSeq 2500 instrument, resulting in approximately 30 million 50 bp reads per sample.

### 2.4. Bioinformatic Analysis

The bioinformatic analysis first mapped sequencing reads to a reference transcriptome, then quantified expression levels of individual genes, and finally identified specific genes that were differentially expressed between VS samples with or without tinnitus. Sequencing reads were mapped in a splice-aware fashion to the human reference transcriptome (hg19/GRCh37.75 assembly) by using the STAR software [37]. Read counts over transcripts were calculated by using HTSeq (a Python package) based on the Ensembl annotation for hg19/GRCh37.75 assembly (http://grch37.ensembl.org/ (accessed on 1 April 2021)) [38]. Principal component analysis was performed to chart the log-fold change (FC) in high-dimensional space (multidimensional scaling) and capture the areas of largest variation. A heat map was generated of the FC in expression to observe clustering of similar expression across groups.

### 2.5. Gene Enrichment Analyses and Gene Set Enrichment Analysis (GSEA)

Functional annotation analysis was performed on differentially expressed genes—defined as those with ≥2-FC and a false discovery rate (FDR) with *p* < 0.05—by using the Database for Annotation, Visualization, and Integrated Discovery (DAVID v6.7; https://david-d.ncifcrf.gov/ (accessed on 17 November 2021)). DAVID is a bioinformatic tool that provides enrichment analysis of gene lists, including Gene Ontology terms [39]. Additionally, functional GSEA was performed on the whole-transcriptome expression values by using a comparison of GSEA24 against Hallmark gene sets with default parameters and a *p*-value cutoff of 0.01 [40]. Hallmark gene sets summarize well-defined biological states or processes and display coherent expression. GSEA first ranked the genes based on a measure of each gene’s differential expression within VS tumors with or without associated tinnitus. Then, the ranked list was used to assess how individual genes within each Hallmark gene set were distributed across the ranked list.

### 2.6. Real-Time Quantitative Reverse Transcription PCR (RT-qPCR)

To validate gene candidates identified by RNA-seq, RNA extracted from 19 FFPE VS samples was reverse-transcribed into cDNA with Taqman Reverse Transcription Reagent (Applied Biosystems, Waltham, MA, USA); three samples were degraded and excluded from all analyses. Real-time RT-qPCR was then performed by using an Applied Biosystems 7700 Sequence Detection System and TaqMan Primers for human *PLVAP*, *GFAP*, *PREX2*, *APLNR*, *APLN*, *KCNQ3*, *GPNMB*, and *NLRP3*, with ribosomal RNA *18S* as the reference gene [41].

### 2.7. Immunohistochemistry

Sectioning and immunostaining of the FFPE tissue blocks were performed by the Dana Farber/Harvard Cancer Center (DF/HCC) Special Histology Core at MGH. Staining was performed for PLVAP (HPA002279, Atlas Antibodies, Bromma, Sweden), PREX2 (HPA015234, Atlas Antibodies), GPNMB (AF2550, R&D Biosystems, Minneapolis, MN, USA), APLNR (ab214369, Abcam, Cambridge, MA, USA), KCNQ3 (ab66640, Abcam), GFAP (Z033429-2, DAKO, Carpinteria, CA), APLN (ab59469, Abcam), NLRP3 (ab214185, Abcam), S100 (PA0900, Leica Biosystems, Wetzlar, Germany), and CD68 (790-2931, Ventana, Oro Valley, AZ, USA).

Immunostaining was performed according to the standard staining protocols of the MGH DF/HCC Special Histology Core. Briefly, this included deparaffinization in xylene, dehydration in ethanol, blocking of endogenous peroxidase activity in hydrogen peroxide diluted in methanol, rinsing in phosphate-buffered saline (PBS), blocking in 10% normal goat serum, incubation with antibody, and washing in PBS. Peripheral nerve tissue was used as a positive control. The intensity of immunostaining was visually ranked by an experienced neuropathologist (ASR) who was blinded to tinnitus status by using a semi-quantitative scale of 0 to 3, where 1 represented light to no staining and 3 represented strong positivity. PLVAP staining was quantified as the number of positive capillaries in three randomly selected visual fields on each slide.

### 2.8. Statistical Analyses

Patients’ demographic and clinical information was individually reported and summarized by group (tinnitus or no tinnitus) using counts and proportions for categorical variables and means with standard deviations (SD) for continuous variables. Age, tumor size, and PTA were compared between groups with unpaired t-tests. WR% was compared between groups with the N-1 chi-squared test. In the comparative analyses of differential gene expression, samples from the group without tinnitus were selected based on a combination of donor patient age, sex, past medical and surgical history, and tumor size in order to be similar to the patient characteristics of samples from the donors with tinnitus.

Differential gene expression was analyzed by using the edgeR method [42]. The criteria for differential expression were defined based on the cutoffs of ≥2-FC in expression levels and FDR < 0.25. Changes in gene expression levels were analyzed by relative quantification and plotted as the fold change by using the 2^−ΔΔCT^ method [43,44]. These data were analyzed with unpaired t-tests with the Holm–Šidák method for multiple comparisons (i.e., adjusted *p*). Nominal *p*-values (i.e., unadjusted *p*) are also reported, but were not adjusted for gene set size. Semi-quantitative scores of the intensity of immunohistochemistry staining were compared by using Mann–Whitney tests with the Holm–Šidák method for multiple comparisons; a *p*-value of 0.05 was used to determine statistical significance. While it is recognized that the p-value is not an optimal methodology for describing statistical associations, we provide these values throughout the manuscript, as some readers may find them useful.

## 3. Results

### 3.1. Characteristics of VS Sample Donors

The demographic and clinical characteristics of donor patients providing VS tissue are summarized in Figure 2 (individual details in Appendix A). Of the 19 total VS samples, 10 were from patients with tinnitus (three females) and nine were from patients without tinnitus (four females). Among the population ultimately included in the RNA-seq (n = 16), there were three females in each group. Patients with and without tinnitus had similar mean (SD) ages (overall: 43.6 [18.6] vs. 49.8 [19.7] years, respectively; RNA-seq sample: 44.1 [19.6] vs. 48.7 [22.5] years) and presurgical tumor sizes (overall: 24.4 [6.2] vs. 30.6 [9.9] cm; RNA-seq sample: 24.8 [6.4] vs. 30.3 [11.1]; all *p* > 0.05). The presurgical PTA and WR% were statistically similar between groups, although patients with tinnitus had numerically worse ipsilateral PTA than those without tinnitus, which approached significance (overall: 79.7 [SD: 39.08] vs. 46.8 [32.02] dB, *p* = 0.09).

### 3.2. RNA-Seq Analysis of Differentially Expressed Genes in VS with or without Associated Tinnitus

Of the 19 VS tissue samples, three did not yield enough high-quality RNA and were excluded from the gene expression analyses, leaving 16 samples for inclusion (n = 9 with tinnitus and n = 7 without tinnitus). Over 1000 genes were differentially expressed between the VS samples with or without associated tinnitus, 11 of which had FDR < 1.0: *RP11-665C14.1*, *CBLN2*, *GFAP*, *CDH7*, *APLNR*, *FOLH1*, *PREX2*, *SHROOM2*, *PLVAP, GPR128*, and *CHRNA1*. The genes that were down-regulated in the tinnitus group, were influenced by more than just an outlier value, and had corresponding FC values of >1 included *GFAP* (logFC= −3.04), *APLNR* (−2.95), *PREX2* (−1.44), and *PLVAP* (−1.04; all *p* < 0.01).

Figure 3A shows a volcano plot of the expression ratios for genes associated with VS samples with tinnitus vs. no tinnitus and the corresponding *p*-values of differential expression, delineating transcripts with ≥2 FC. The heatmap in Figure 3B shows the expression values of 179 differentially expressed genes with >2 FC in expression and *p* < 0.01, clustered hierarchically by columns (samples) using 1-Spearman rank correlation and linked by averages.

### 3.3. Functional Enrichment Analysis and GSEA

Trends in differential gene expression were further explored via functional enrichment analysis, which revealed several protein categories that were significantly enriched in VS patients with tinnitus, including glycoproteins, signal peptides, and secreted proteins (Figure 4A). Three candidate genes (*GPNMB*, *PLVAP*, and *APLNR*) were characterized as glycoproteins. *APLNR* was also classified in the ‘receptor’ and ‘neuroactive ligand-receptor interaction’ categories.

Further analysis of the 16 VS samples that underwent RNA-seq was performed with GSEA to test whether sets of related genes were systematically altered between the two groups. The top pathways for the VS samples with tinnitus were allograft rejection, pancreas beta cells, tumor necrosis factor (TNF) signaling, complement, multiple interleukin signaling pathways, xenobolic metabolism, p53, and bile acid metabolism (Figure 4B, Table 1). Of these, five were associated with inflammation and included the genes *TNF* and *INFG*. The top pathway in this GSEA was related to allograft rejection and was significantly enriched in VS samples associated with tinnitus (FDR = 0.022; nominal *p* < 0.001; family-wise error rate = 0.045). This pathway included the following genes: matrix metallopeptidase 9 (*MMP9)*, C-X-C motif chemokine ligand 9 (*CXCL9)*, interleukin 16 (*IL16)*, platelet factor 4 (*PF4)*, IL2 inducible T cell kinase (*ITK)*, and activin A receptor type 2A (*ACVR2A*).

### 3.4. Validation of RNA-Seq Results with Real-Time RT-qPCR and Immunohistochemistry

To validate the RNA-seq results, RNA extracted from the 16 VS tissue samples was used for real-time RT-qPCR of eight genes of interest: *PREX2*, *APLNR*, *GFAP*, *PLVAP,* and *GPNMB,* which were identified above, *APLN* because it is a ligand for *APLNR* [45] and because the plasma levels of APLN were reported to be negatively correlated with the severity of bilateral idiopathic tinnitus [46], *KCNQ3* because this potassium channel is implicated in noise-induced tinnitus [47], and *NLRP3* because NLRP3 inflammasome activation is implicated in VS-induced hearing loss [41]. All genes except *PLVAP* were detected with real-time RT-qPCR in both groups (tinnitus and no tinnitus). While the trends in RT-qPCR gene expression were consistent with RNA-seq data, they did not meet our criterion for significance (Figure 5).

All 19 VS tinnitus samples were examined with immunohistochemistry for KCNQ3, NLRP3, PREX2, GFAP, GPNMB, APLNR, and APLN, as well as S100 (a Schwann cell marker) and CD68 (a macrophage and monocyte marker) (see representative images in Figure 6A). While the difference in the semi-quantitative scoring of staining intensity between samples with and without associated tinnitus did not meet our criterion for significance, there were notable trends (Figure 6B). For example, there was a trend toward greater staining for KCNQ3, NLRP3, and CD68 and lower staining for PREX2 and APLN in the VS samples associated with tinnitus compared to those without tinnitus. These trends were directionally consistent with the results of the RNA-seq analysis with regard to the lower PREX2 and APLN in VS samples associated with tinnitus. PLVAP staining was limited to capillaries, and no significant differences in PLVAP staining or the number of capillaries were observed between the VS groups (Figure 6C).

## 4. Discussion

To the best of our knowledge, this is the first study to compare differential gene expression in archival VS tissue from patients with and without presurgical tinnitus. Other than the diagnosis of tinnitus, the two groups were generally alike in terms of their demographic and clinical characteristics, allowing comparisons of VS tumor gene expression in a similar patient population. By using next-generation sequencing on RNA extracted from FFPE VS tissue, bioinformatic analysis, and GSEA, we identified novel and previously reported molecular pathways and genes that are potentially associated with tinnitus.

In the GSEA, the sole Hallmark pathway that was significantly enriched in VS tissue associated with tinnitus in comparison with that without tinnitus was the allograft rejection pathway. This pathway is involved in the adaptive immune response for allograft destruction, wherein antigen-presenting cells activate naïve T-cells, leading to CD8+ and CD4+ T cell maturation [48]. This ultimately results in the destruction of allograft cells by apoptosis via CD8+ activated T-cells or by TNF-α and free-radical-mediated cytotoxicity via CD4+ activated T-cells (which differentiate into Type 1 helper [Th1] cells). Th1 cells also produce cytokines such as interferon-gamma, various interleukins, and TNF-β, which activate macrophages.

This finding is intriguing, as it provides additional evidence that the local immune environment of a VS tumor may play a role in the development of tinnitus, perhaps through the aberrant destruction of cochlear cells or auditory neurons (peripheral or central) by immune cells activated by or recruited to the tumor. There is accumulating evidence that tinnitus is associated with neuroinflammation [49], such as a positive correlation between the circulating cytokines IL-1β and TNF-α with psychometric scores of tinnitus [50]. In noise-exposed rodents, tinnitus was associated with increased pro-inflammatory cytokines and microglial activation in the primary auditory cortex [51]. Importantly, the *NF2* gene, which is mutated in VS tumors, normally encodes for Merlin, a potent inhibitor of the immune-related NF-κB pathway [52]. NF-κB has a known role in allograft rejection—it is activated in the graft within a few hours due to ischemia and reperfusion, and then again as immune cells infiltrate [53]. Cytokines such as TNF-α prompt transcription of NF-κB in an autoregulatory positive feedback loop, as NF-κB activation itself upregulates TNF-α, in addition to other cytokines [54]. TNF-α is known to be both ototoxic and secreted by human VS cells [55,56]. Thus, the loss of or decrease in Merlin in VS may promote dysfunctional activation of NF-κB, which is supported by a network analysis of genes expressed in VS tumors that identified NF-κB as the lynchpin of differentially up-regulated immune-related genes [57]. Prior studies have suggested an ototoxic role for TNF-α, which may be tied to the allograft rejection pathway. In mice, the application of recombinant TNF-α to neonatal cochlear explants caused ototoxicity manifesting as neurite disorganization and hair cell loss [56]. Similarly, cochlear perfusion of TNF-α in guinea pigs resulted in hearing loss and cochlear synaptic degeneration [58], as well as a reduction in cochlear blood flow [59]. Additionally, infusion of TNF-α led to a tinnitus phenotype in mice, while knockdown or inhibition of TNF-α, as well as microglial depletion, prevented tinnitus [51].

The top differentially expressed gene in the allograft pathway, *CXCL9*, is of particular interest, as the expression of the chemokine CXCL9 boosts T-cell tumor infiltration and inhibits tumor growth, angiogenesis, and metastasis [60,61]. CXCL9 is part of an axis regulating immune cell migration, differentiation, and activation. It is expressed at low levels under homeostatic conditions but is up-regulated during cytokine stimulation [62]. This study observed higher expression of *CXCL9* in the group with tinnitus; thus, it could be hypothesized that the cytokine activation occurring in these VS tumors elevated *CXCL9* expression. This may partially explain the observation that the VS tumors from patients without tinnitus were, on average, numerically smaller than those of patients without tinnitus. Transcriptional regulation of CXCL9 is complex, involving both the signal transducer and activator of transcription (STAT1) and NF-κB [63]. In other solid tumors (e.g., ovarian or breast), CXCL9 is a favorable prognostic biomarker that indicates that the cancer will respond to immune checkpoint inhibitor therapy and is associated with longer survival in some cases [64,65]. Future studies are needed to further delineate the role of CXCL9 in VS tumors and whether it could be used as a similar prognostic factor for tinnitus.

A surprising finding from the RNA-seq analysis was that *PREX2* expression was lower in VS tissue from patients with tinnitus compared to those without tinnitus, which was directionally validated with real-time RT-qPCR and immunohistochemistry. PREX2 regulates the exchange of GTP on Rac1, and it binds to and inhibits the tumor suppressor PTEN (which can, likewise, inhibit PREX2) [66,67]. Accordingly, *PREX2* is widely considered an oncogene due to its promotion of tumor proliferation, migration, and invasion in numerous neoplasms (i.e., glioma, melanoma, and breast, ovarian, prostatic, and pancreatic cancers) when mutated or overexpressed [68,69,70,71]. In patients with NF2-associated schwannoma (who typically have bilateral VS), down-regulation of Rac1 activity inhibited proliferation and induced apoptosis in schwannoma [72]. *PTEN* has been implicated as a contributor (although not the primary catalyst) of VS tumorigenesis [73,74], and it is heterogeneously expressed across sporadic VS tumors (i.e., in about 70%) [75]. Interestingly, the VS patients with tinnitus in this study had, in addition to lower expression of *PREX2,* numerically smaller tumors than those without tinnitus. It is possible that increased *PREX2* expression contributes to a pro-tumorigenic environment in VS, similarly to that in other cancers, which may have secondary ototoxic effects that are unrelated to size alone. Up-regulation of PREX2 is considered a prognostic factor for poorer outcomes for patients with breast and prostate cancer [67]. Further studies are needed to test if there is a relationship between *PREX2* expression and adverse VS-specific patient outcomes, such as tinnitus.

Several other genes were found to be differentially expressed in VS tumors with vs. without tinnitus in the RNA-seq analysis, including *APLNR* and *PLVAP.* The role of the neuropeptide apelin and its receptor is yet unknown, although it is hypothesized to have neuroprotective effects against oxidative stress. Apelin expression was lower in VS tumors with tinnitus than it was in those without tinnitus. Immunopositive apelin conglomerates did not overlap with macrophage/monocyte marker CD68-positive or Schwann cell marker S100-positive immunostaining on consecutive slides. Previous studies observed lower mean plasma levels of apelin in patients with tinnitus than in those without tinnitus [46]. Additionally, noise-exposed rats that were administered apelin-13 showed attenuated hearing loss compared to controls, which was hypothesized to be via sirtuin-1 regulation [76]. Apelin demonstrated a protective effect against cisplatin-induced ototoxicity in vitro via the inhibition of reactive oxygen species and apoptosis [77]. The apelin receptor *APLNR* was also found to be down-regulated in VS versus control nerves, although tinnitus, hearing loss, and tumor size were not separately examined [78]. 

The results of this study should be considered in light of several limitations, some of which are inherent to the study of rare diseases in humans. For example, the sample size of this study was generally small (N = 19 VS tumor samples) due to the rarity of VS, the even greater rarity of VS patients without tinnitus to serve as controls for VS patients with tinnitus, and the difficulty in procuring human VS tissue for research. The sample size was further restricted in the RNA-seq analysis due to the low-quality RNA from three samples. This may have impacted the ability to detect significant differences in gene expression between groups, although the directionality of the expression of candidate genes was generally confirmed in the RT-qPCR and immunohistochemistry results. Additionally, two patients had NF2-associated VS, which may have resulted in gene expression that differed from that in patients with sporadic VS. Furthermore, it was challenging to determine whether or not patients had tinnitus preoperatively given the lack of objective tinnitus measurements; therefore, subjective documentation from clinicians and audiologists was used.

Due in part to the diverse etiology of tinnitus, the lack of objective markers for this subjective condition, and our presently poor understanding of its pathophysiology, there are currently no effective pharmaceutical therapies targeting its cause [79]. Furthermore, the limited available treatments, such as sound masking, hearing aids, or management of distressing symptoms with behavioral therapy, are associated with heterogenous efficacy and patient response. Several novel therapies, including the use of cochlear implants (CIs) and repetitive transcranial magnetic stimulation, have shown promise for tinnitus, although they remain experimental and are not routinely used in clinical practice [80,81]. CIs, which are typically reserved for patients with severe to profound sensorineural hearing loss, can alleviate tinnitus in 54–86% of patients without VS [82,83]. Cochlear implantation is a relatively new indication for hearing restoration in some VS patients and is relevant when the cochlear nerve is preserved during microsurgical tumor resection, following stereotactic radiation surgery, or when tumor resection is not attempted due to the tumor’s small size and the patient’s overall medical condition [84]. A retrospective cohort study of VS patients with tinnitus reported that median Tinnitus Handicap Inventory scores significantly improved after CI implantation (n = 17) [80], which is consistent with several prior small cohort studies that also reported improvements in tinnitus symptoms with CIs [85,86,87]. However, VS tumor management and the tumors themselves may have unpredictable effects on CIs’ function given the damage to the vestibulocochlear nerve. Thus, novel non-invasive therapies are still urgently needed. As tinnitus is associated with a reduced quality of life, particularly for those with chronic bothersome symptoms, in addition to a staggering global economic burden [88,89,90,91,92], it is imperative to improve our understanding of tinnitus at the molecular level. The present results contribute insight into potential genes and pathways associated with tinnitus in VS that should be further validated in larger cohorts.

## 5. Conclusions

This next-generation sequencing and bioinformatic analysis revealed that the allograft rejection pathway, an inflammatory pathway, and associated genes were differentially expressed in VS tissue with and without associated tinnitus. Future studies are recommended to further explore the role of inflammation and the candidate genes identified herein in VS-associated tinnitus.

## Figures and Tables

**Figure 1 jcm-12-02642-f001:**
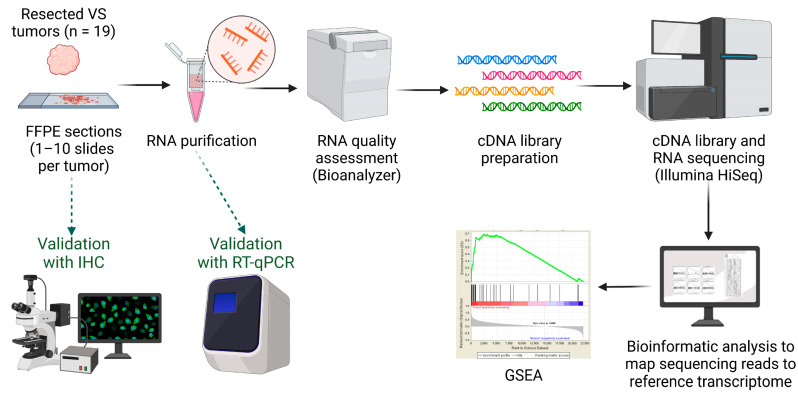
Methodological approach. Abbreviations: cDNA, complementary deoxyribonucleic acid; FFPE, formalin-fixed paraffin-embedded; GSEA, gene set enrichment analysis; IHC, immunohistochemistry; RNA, ribonucleic acid; RT-qPCR, reverse transcription–quantitative polymerase chain reaction; VS, vestibular schwannoma. Created with BioRender (www.biorender.com, accessed on 24 March 2023).

**Figure 2 jcm-12-02642-f002:**
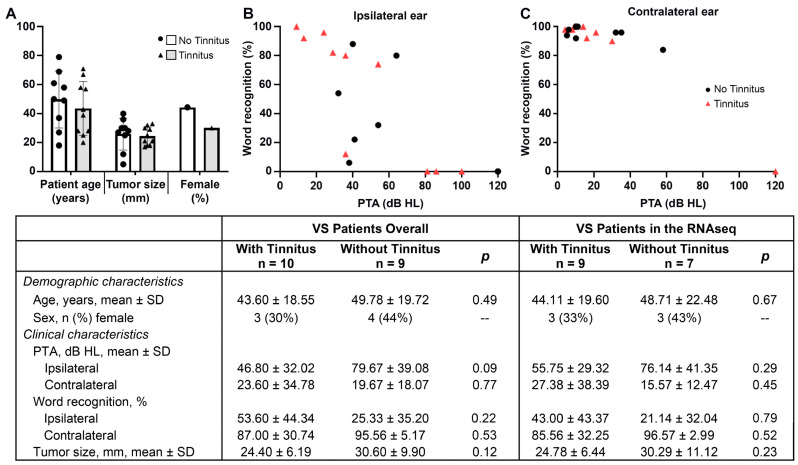
Demographic and clinical characteristics of VS tissue donors with and without tinnitus. The distribution of patient age, sex, and tumor size are shown in (**A**), while patients’ pre-surgical PTA ipsilateral and contralateral to the tumor are in (**B**,**C**), respectively. The clinical and demographic characteristics were not significantly different between the included patients with tinnitus (triangles, gray bars) or without tinnitus (black dots, white bars) overall or among patients included in the RNA sequencing. Word recognition, PTA, and presurgical tumor size (measured in transverse dimension in the cerebellopontine angle via magnetic resonance imaging) were collected from the most recent test/scan prior to VS tumor resection. Abbreviations: dB, decibel; HL, hearing level; PTA, pure tone average; SD, standard deviation.

**Figure 3 jcm-12-02642-f003:**
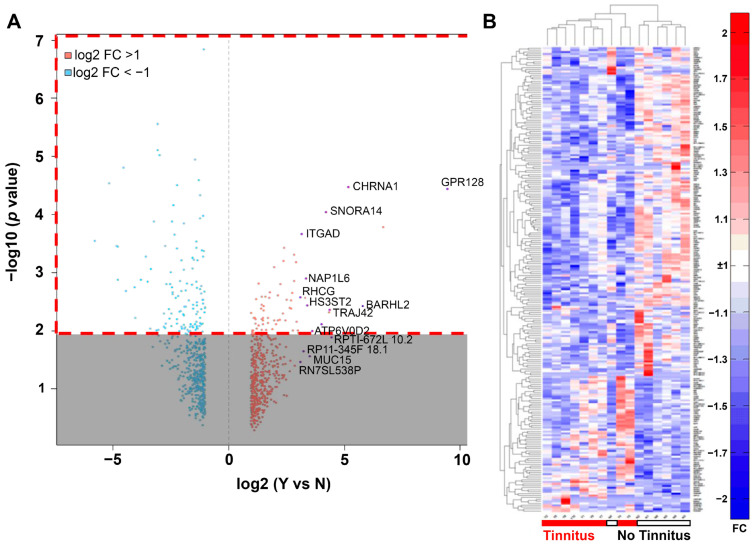
Bioinformatic analysis of genes that were differentially expressed in VS samples stratified by the presence or absence of associated tinnitus. (**A**) Volcano plot highlighting differentially expressed genes associated with tinnitus (n = 9 [VS samples]) vs. no tinnitus (n = 7). Transcripts with >2-fold change in expression are shown as red (up-regulated) and blue dots (down-regulated). The dashed red box highlights genes that were selected for the heatmap. (**B**) Heatmap of RNA-seq gene expression levels (log2 of RPKM) compared between patients with and without tinnitus. A total of 179 differentially expressed genes were detected based on the cutoffs of >2-fold change and *p*-value < 0.01. The heatmap is clustered hierarchically by both samples (columns) and genes (rows). Red indicates differentially higher and blue indicates differentially lower levels of gene expression between groups. Abbreviations: RNA-seq, RNA sequencing; VS, vestibular schwannoma.

**Figure 4 jcm-12-02642-f004:**
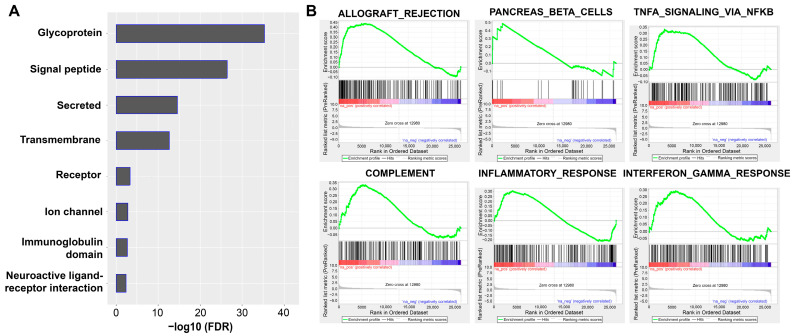
Enrichment analysis of differentially expressed genes in VS stratified by the presence or absence of tinnitus. (**A**) Functional enrichment analysis of differentially expressed genes by using DAVID revealed functional gene categories. (**B**) All genes ranked according to the magnitude of expression change (fold change) between VS patients with and without tinnitus were analyzed for the enrichment of functional gene categories by using GSEA. Five of the six top GSEAs were associated with inflammation. The green line indicates the enrichment score. Red indicates positive correlation and blue indicates negative correlation in the ranked ordered dataset.

**Figure 5 jcm-12-02642-f005:**
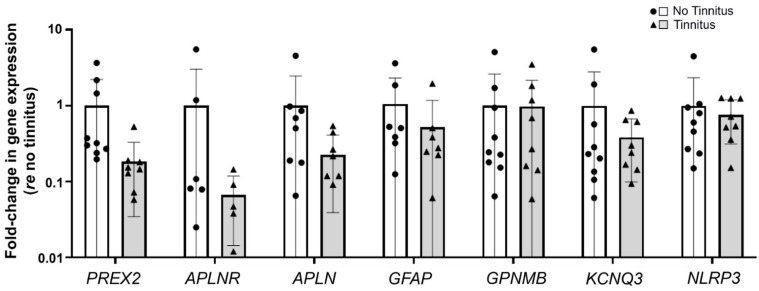
RT-qPCR validation of candidate genes identified via RNA-seq or implicated in previous tinnitus studies. VS tissue from patients without tinnitus (white bars, circles) was compared to VS tissue from patients with tinnitus (gray bar, triangles) (n = 19 total samples; see Appendix A for patient information). While there was a trend for lower expression of several genes in VS tissue associated with tinnitus, this trend did not meet our criterion for significance. Error bars indicate standard deviations. Abbreviations: RNA-seq, RNA sequencing; RT-qPCR, reverse transcription–quantitative polymerase chain reaction.

**Figure 6 jcm-12-02642-f006:**
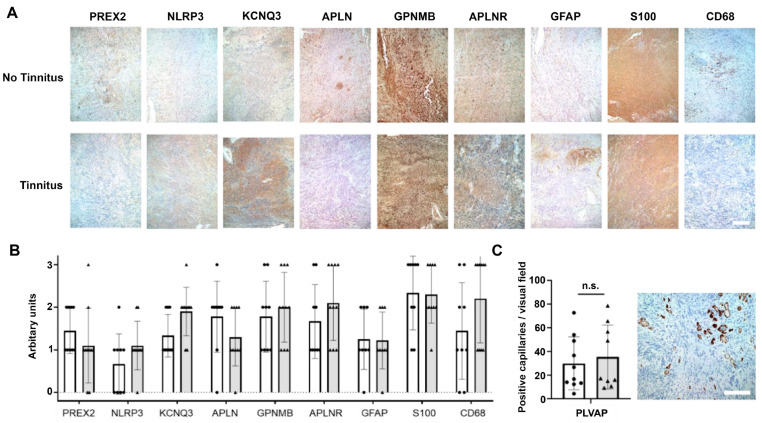
Validation of candidate genes identified via RNA-seq with immunohistochemistry. (**A**) Representative immunohistochemistry images of VS tissue from one patient without tinnitus (VS.N6) and one patient with tinnitus (VS.Y8). Tissue sections were immunostained for KCNQ3, NLRP3, PREX2, GFAP, GPNMB, APLNR, APLN, S100, and CD68. Scale bar = 200 µm. (**B**) Semi-quantitative measure of staining intensity in VS samples from patients with (gray bars, triangles) or without (white bars, circles) tinnitus. The scoring ranges from 1 (light to no staining) to 3 (strong positivity). There were no significant differences in staining intensity between the groups. (**C**) PLVAP staining of capillaries in VS tissue (example from VS-198). There was no significant difference in PLVAP staining intensity or number of capillaries between the samples from patients with (gray bars and triangles) or without (white bars and circles) tinnitus. Error bars indicate the SD in panels B and C. Abbreviations: n.s., not significant; SD, standard deviation; VS, vestibular schwannoma.

**Table 1 jcm-12-02642-t001:** Functional pathways enriched in VS tumor tissue from patients with tinnitus compared with those without tinnitus according to GSEA.

		*p*-Values		
Pathway	Description	Nominal ^a^	FDR ^b^	FWER
HALLMARK_ALLOGRAFT_REJECTION	Genes up-regulated during transplant rejection	0 *	0.022 *	0.045 *
HALLMARK_PANCREAS_BETA_CELLS	Genes specifically up-regulated in pancreatic beta cells	0.134	0.309	0.741
HALLMARK_TNFA_SIGNALING_VIA_NFKB	Genes regulated by NF-kB in response to TNFα	0.04	0.284	0.845
HALLMARK_COMPLEMENT	Genes encoding components of the complement system, which is part of the innate immune system	0.070	0.269	0.898
HALLMARK_INFLAMMATORY_RESPONSE	Genes defining inflammatory response	0.114	0.353	0.979
HALLMARK_INTERFERON_GAMMA_RESPONSE	Genes up-regulated in response to IFNG	0.222	0.491	0.998

Five of the top six enriched pathways were associated with inflammation. Only the enrichment level within the HALLMARK_ALLOGRAFT_REJECTION pathway was significantly different between VS samples from patients with and without tinnitus (* FDR < 0.25, nominal and FWER *p* < 0.05). Abbreviations: GSEA, gene set enrichment analysis; FDR, false discovery rate; FWER, family-wise error rate; IFNG, interferon-gamma; NF-kB, nuclear factor kappa-light-chain enhancer of activated B cells; TNF-α, tumor necrosis factor alpha. Note: ^a^ Unadjusted *p*-value. ^b^ Adjusted *p*-value for gene set size and multiple hypothesis testing.

## Data Availability

The authors attest that all study data are included in the article or online supplemental material.

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
