# Peer review of "Comparative Transcriptomic Analysis of Archival Human Vestibular Schwannoma Tissue from Patients with and without Tinnitus"

_jcm, 2023, doi:10.3390/jcm12072642_

Round 1
Reviewer 1 Report
Dear Authors/Editors
I find the article approach quite interesting, hopefully more information on the origins of tinnitus will be available following this line of research.
The citation of improovement of tinnitus with surgery or cochlear implants would be interesting, but overall I like the article very much.
Reviewer 2 Report
REVIEW REPORT
This is a cutting-edge Bioinformatics research, in which the researchers analyzed differentially-expressed genes in tissue sections of 9 cases of Vestibular schwannoma (VS)with tinnitus (VS-Tin) and control of 7 patients without tinnitus (VS-NoTin). Using RNA sequence analysis to elucidate the transcription of genes, they found that VS-Tin had significantly lower expression of GFAP, APLNR, PREX2, and PLVAP genes. This transcription information was validated using real-time RT-qPCR. Their Immuno-histochemistry biopsy analysis revealed inflammatory markers were activated in VS-Tin group, namely NLRP3 inflammasome and CD68-positive macrophages. Reduced expression of less PREX2 and apelin expression were also noted. Functional enrichment analysis found glycoproteins, signal peptides, and secreted proteins were enhanced in study group VS-Tin and gene set enrichment analysis showed that inflammatory pathway was actuated.
This is a praiseworthy and groundbreaking high quality research work. The study has ethical committee approval of Massachusetts General Hospital and Massachusetts Eye & Ear (Protocol # 14-148H) which further validates this research and participant informed consent protocol was followed. In addition study data is available online for scrutiny. The study highlights the opening up of allo-graft rejection pathway, and inflammatory pathway in Vestibular schwannoma (VS)with tinnitus.
Small sample size of (N=9/7) is the limitation of this research. Further multiple analyses are clubbed into a single group. But these limitations are tolerable in a high-quality study of such nature as the Vestibular schwannoma disease itself is not very common, and not every patient has tinnitus or related symptoms, and thus smaller sample size maybe acceptable.
Reviewer 3 Report
The manuscript presents a prototypic comparative transcriptomic analysis of archival human vestibular schwannoma(VS) tissue from patients with or without tinnitus. The authors applied next-generation sequencing, used bioinformatic analysis to detect differentially-expressed genes, immunohistochemistry staining as the gold standard revealed a trend for less PREX2 and apelin expression and greater expression of NLRP3 inflammasome and CD68-positive macrophage in VS-Tin vs VS-No Tin, suggesting an inflammatory pathway activation in VS-Tin. As recently known inflammation and oxidative stress represent the main pathways of endothelial dysfunction. Three of 19 tissue samples did not yield enough high-quality RNA and were excluded from the gene expression analyses, leaving 16 samples for inclusion(9 with tinnitus and 7 without tinnitus).
While there was a trend for lower expression of several genes in VS tissue associated with tinnitus, genes down-regulated in the tinnitus group, influenced by more than just an outlier value, this trend did not meet treir criteria for significance.
I'd like to see an increased sample size of more than 20 with tinnitus(present sample is 9)and better (as possible) determine whether patients had tinnitus preoperatively given the lack of objective measurements, in order to not (or less) influence these important results from the precedent minor limitations, and strengthen the model with rewarding and sealing the results, blazing to become a reference point and state of the art article.
